

# BengSentiLex and BengSwearLex: creating lexicons for sentiment analysis and profanity detection in low-resource Bengali language

Salim Sazzed

Department of Computer Science, Old Dominion University, Norfolk, VA, USA

## ABSTRACT

Bengali is a low-resource language that lacks tools and resources for various natural language processing (NLP) tasks, such as sentiment analysis or profanity identification. In Bengali, only the translated versions of English sentiment lexicons are available. Moreover, no dictionary exists for detecting profanity in Bengali social media text. This study introduces a Bengali sentiment lexicon, BengSentiLex, and a Bengali swear lexicon, BengSwearLex. For creating BengSentiLex, a cross-lingual methodology is proposed that utilizes a machine translation system, a review corpus, two English sentiment lexicons, pointwise mutual information (PMI), and supervised machine learning (ML) classifiers in various stages. A semi-automatic methodology is presented to develop BengSwearLex that leverages an obscene corpus, word embedding, and part-of-speech (POS) taggers. The performance of BengSentiLex compared with the translated English lexicons in three evaluation datasets. BengSentiLex achieves 5%–50% improvement over the translated lexicons. For identifying profanity, BengSwearLex achieves documentlevel coverage of around 85% in an document-level in the evaluation dataset. The experimental results imply that BengSentiLex and BengSwearLex are effective resources for classifying sentiment and identifying profanity in Bengali social media content, respectively.

# INTRODUCTION

The popularity of e-commerce and social media has surged the availability of user-generated content. Therefore, text analysis tasks such as sentiment classification and inappropriate content identification, have received significant attention in recent years. Sentiment analysis identifies emotions, attitudes, and opinions expressed in a text (*Liu, 2012*). Extracting insights from user feedback data has practical implications for market research, customer service, result predictions, *etc.* Profanity indicates the use of taboo or swear words to express emotional feelings (*Wang et al., 2014*) and is prevalent in various types of social media data (*e.g.*, online post, message, comment, *etc.*) across languages. The occurrences of swearing or vulgar words are often linked with abusive or hatred context, sexism, and racism. Hence, identifying swearing words has practical connections to understanding and

Corresponding author
Salim Sazzed, ssazz001@odu.edu

monitoring online content. In this paper, we use terms such as profanity, slang, vulgarity, and swearing interchangeably to indicate the usage of foul and filthy language even though they have subtle differences in their meanings.

Lexicon plays a significant role in both sentiment classification and profanity identification. For example, sentiment lexicons help to analyze key subjective properties of texts, such as opinions and attitudes (*Taboada et al., 2011*). A sentiment lexicon contains opinion conveying terms (*e.g.*, words, phrases, *etc.*), labeled with the sentiment polarity (*e.g., positive or negative*) and polarity strength. Positive sentiment words such as *beautiful and good* express some desired states or qualities, while negative sentiment words, such as *badand and awful* represent undesired states. The profane list, on the other hand, contains words having foul, filthy, and profane meanings (*e.g., ass, fuck, bitch*). When labeled data are unavailable, sentiment classification methods usually utilize opinion conveying words and a set of linguistic rules. As this approach relies on the polarity of the individual words, it is crucial to building a comprehensive sentiment lexicon. Similarly, creating a lexicon that consists of a list of swear and obscene words is instrumental for determining profanity in a text.

As sentiment analysis is a well-studied problem in English, many general-purpose and domain-specific 49 sentiment lexicons are available, such as MPQI (*Wilson, Wiebe & Hoffmann, 2005*), opinion lexicon (*Hu & Liu, 2004*), SentiwordNet (*Esuli & Sebastiani, 2006*), VADER (*Hutto & Gilbert, 2014*), *etc.* Besides English, other widely used languages such as Chinese, Arabic, and Spanish have their sentiment lexicons (*Xu, Meng & Wang, 2010*; *Mohammad, Salameh & Kiritchenko, 2016*; *Perez-Rosas, Banea & Mihalcea, 2012*). The presence of swearing in English social media has been investigated by various researchers (*Wang et al., 2014*; *Pamungkas, Basile & Patti, 2020*). *Wang et al. (2014)* found that the rate of swear word use in English Twitter is 1.15%, almost double compared to its use in daily conversation (0.5% - 0.7%) as observed in previous work (*Jay, 1992*). The work of *Wang et al. (2014)* also reported that a portion of 7.73% tweets in their random sampling collection contains swear words.

Although Bengali is the seventh most spoken language in the world, NLP tasks such as sentiment analysis or profanity identification have not been fully explored in Bengali yet. In the last few decades, 59 limited research has been conducted on sentiment analysis in Bengali mostly by utilizing supervised machine learning (ML) classifiers (*Sazzed & Jayarathna, 2019*; *Das & Bandyopadhyay, 2010a*; *Chowdhury & Chowdhury, 2014*; *Sarkar & Bhowmick, 2017*; *Rahman & Kumar Dey, 2018a*), ML classifiers do not requir require language-specific resources such as sentiment lexicon, part-of-speech (POS) tagger, or dependency parser, *etc.* Regarding profanity identification, although few works addressed the abusive content analysis, none of them focused on determining profanity or generating resources for identifying profanity.

There have been a few attempts to develop sentiment lexicons for Bengali by translating various English sentiment dictionaries. *Das & Bandyopadhyay (2010b)* utilized a word-level lexical-transfer technique and an English-Bengali dictionary to develop SentiWordNet for Bengali from English SentiWordNet. *Amin et al. (2019)* translated the VADER (*Hutto & Gilbert, 2014*) sentiment lexicon to Bengali for sentiment analysis.

However, dictionary-based translation can not capture the informal language people use in everyday communication or in social media. Besides, there exists no obscene lexicon to identify profanity in Bengali social media text data. Therefore, in this work, we focus on generating resources for these two essential tasks.

To develop the Bengali sentiment lexicon BengSentiLex, we present a corpus-based cross-lingual methodology. The proposed framework leverages around 50,000 Bengali drama reviews from Youtube; among them, we manually annotated around 12,000 reviews; among them, 11,807 reviews are labeled (*Sazzed, 2020a*) while the remaining reviews are unlabeled. The proposed framework consists of three phases, where each phase identifies sentiment words from the drama review corpus and includes them in the lexicon. In phase 1, we identify sentiment words from both labeled and unlabeled Bengali reviews with the help of of two English sentiment lexicons, opinion lexicon 80 (*Hu & Liu, 2004*) and VADER (*Hutto & Gilbert, 2014*). In phase 2, utilizing around 12,000 annotated reviews and PMI, we identify top class-correlated (*positive* or *negative*) words. Using the POS tagger, we determine adjectives and verbs, which mainly convey opinions. In the final phase, we make use of unlabeled reviews to recognize the polar words. Utilizing the labeled reviews as training data, we determine the class of the unlabeled reviews. We then follow the similar steps of phase 2 to identify sentiment words from these pseudo-labeled reviews. All three phases are followed by a manual validation and synonym generation step. Finally, we provide the comparative performance analysis of the developed BengSentiLex with the translated English lexicons. As BengSentiLex is built from a social media corpus, it contains words that people use on the web, social media, and informal communication; Therefore, it is more effective in recognizing sentiments in text data compared to word-level translation of English lexicons. The earlier version of BengSentiLex has been discussed in *Sazzed (2020b)*.

To construct the Bengali swear lexicon, BengSwearLex, we propose a corpus-based semi-automatic approach. Unlike the framework of BengSentiLex, this approach does not leverage any cross-lingual resources as machine translation is not capable of translating language-specific swear or obscene terms. From an existing Bengali obscene corpus, utilizing word embedding and POS tagging, we create BengSwearLex. To show the efficacy of BengSwearLex for identifying profanity, we annotate a negative drama review corpus into profane and non-profane categories based on the presence of swear terms. We find that BengSwearLex successfully identifies 85.5% of the profane reviews from the corpus.

## Motivation and challenges

Since the existing Bengali sentiment dictionaries lack words people use in informal and social communication, it is necessary to build such a sentiment lexicon in Bengali. With the rapid growth of user-generated Bengali content on social media and the web, the presence of inappropriate content has become an issue. The content which is not in line with the social norms and expectations of a community needs to be censored. Unfortunately, no such resources exist for identifying obscene or profane textual content in Bengali social media. Therefore, this study strives to build a swear lexicon for Bengali.

Some of the challenges to develop a sentiment lexicon in Bengali are-

1. One of the popular techniques to create a lexicon is to utilize corpora to extract opinion conveying words. Since the Bengali language lacks such a corpus. Thus, we have to collect and annotate a large corpus.
2. One of the necessary tools for identifying opinion word is a sophisticated part-of-speech (POS) tagger; However, in Bengali, there exists no sophisticated POS tagger exists; thus, we leverage POS tagger from English utilizing machine translation.

### Contributions

The main contributions of this paper can be summarized as follows-

- We introduce two lexical resources, the BengSentiLex, a Bengali sentiment lexicon consisting of over 1,200 opinion words created from a Bengali review corpus, and BengSwearLex, a Bengali swear lexicon, comprised of about 200 swear words. We make both resources publicly available for the researchers.
- We show thet how the machine translation-based cross-lingual approach, the labeled and unlabeled reviews, and English sentiment lexicons can be utilized to build a sentiment lexicon in Bengali.
- We present a semi-automatic methodology for developing a swear lexicon utilizing an obscene corpus and various natural language processing tools.
- We demonstrate that BengSentiLex and BengSwearLex are effective at sentiment classification and profane terms detection, respectively, compared to existing tools.

## WORK RELATED TO SENTIMENT LEXICON CREATION

*Liu (2012)* categorized the sentiment lexicon generation methods into three categories, manual approach, dictionary-based approach, and corpus-based approach. Considerable time and resources are required in the manual approach as the annotation process is performed by individuals. The dictionary-based methods usually start with a set of manually created seed words. In the subsequent steps, seed words are expanded using a dictionary. The corpus-based techniques utilize both manually labeled seed words and corpus data. In this section, we discuss only the related corpus-based lexicon creation methods since BengSentiLex is a corpus-based approach.

### Corpus-based lexicon generation in English

*Huang, Niu & Shi (2014)* proposed a label propagation-based method for generating domain-specific sentiment lexicon. The authors extracted candidate sentiment terms are extracted by leveraging the chunk dependency information and prior generic sentiment dictionary. They defined the pairwise contextual and morphological constraints and incorporated the label propagation. Their experimental results suggested that constrained label propagation can improve the performance of the automatic construction of domain-specific sentiment lexicon.

*Han et al. (2018)* proposed a domain-specific lexicon generation method from the unlabeled corpus utilizing mutual information and part-of-speech (POS) tags. Their lexicon shows satisfactory performance on several publicly available datasets. *Tai & Kao (2013)* proposed a graph-based label propagation algorithm that considers words as nodes

and similarities as weighted edges of the word graphs. Using a graph-based label propagation method, they assigned the polarity to unlabeled words. They conducted experiments on the Twitter dataset and achieved better performance than the general-purpose sentiment dictionaries.

*Wang & Xia (2017)* developed a neural architecture to train a sentiment-aware word embedding. For improving th quality of word embedding and sentiment lexicon, they integrated the sentiment supervision at both document and word levels. They performed experiments on the SemEval 2013-2016 datasets using their sentiment lexicon and obtained the best performance in both supervised and unsupervised sentiment classification tasks.

*Hamilton et al. (2016)* constructed a domain-sensitive sentiment lexicon using label propagation algorithms and small seed sets. They showed that their corpus-based approach outperformed methods that rely on hand-curated resources such as WordNet.

*Wu et al. (2019)* presented an automatic method for building a target-specific sentiment lexicon. Their lexicon consists of opinion pairs made from an opinion target and an opinion word. Their unsupervised algorithms first extract high-quality opinion pairs; Then, utilizing general-purpose sentiment lexicon and contextual knowledge, it calculates sentiment scores of opinion pairs. They applied their method on several product review datasets and found their lexicon outperformed several general-purpose sentiment lexicons.

*Beigi & Moattar (2021)* presented a domain-specific automatic sentiment lexicon construction method for unsupervised domain adaptation and sentiment classification. The authors first constructed a sentiment lexicon from the source domain using the labeled data. In the next phase, the weights of the first hidden layer of Multilayer Perceptron (MLP) were set to the corresponding polarity score of each word from the developed sentiment lexicon, and then the network is trained. Finally, the Domain-independent Lexicon (DIL) is introduced that contains words with static positive or negative scores. The experiments on Amazon multi-domain sentiment datasets showed showed the advantages of their approach over the existing unsupervised domain adaptation methods.

## Lexicon generation in Bengali and other non-English languages

*Al-Moslmi et al. (2018)* developed an Arabic sentiment lexicon consisting of 3880 positive and negative synsets annotated with the part-of-speech (POS) tags, polarity scores, dialects synsets, and inflected forms. The authors performed the word-level translation of the English MPQA lexicon using google translation and then manually inspected it to remove the inappropriate word. Besides, they manually examined a list of opinion and sentiment words and phrases from two Arabic review corpora.

*Perez-Rosas, Banea & Mihalcea (2012)*, the authors presented a framework to derive sentiment lexicon in Spanish using manually and automatically annotated data from English. To bridge the language gap, they used the multilingual sense-level aligned WordNet structure. *Mohammad, Salameh & Kiritchenko (2016)* generated several sentiment lexicons in Arabic using two different methods: (1) by using distant supervision techniques on Arabic tweets, and (2) by translating English sentiment lexicons into Arabic using a freely available statistical machine translation system. They compared the performance of existing and their proposed sentiment lexicons in sentence-level sentiment analysis. *Asghar et al.*

*(2019)* presented a word-level translation scheme for creating an Urdu polarity lexicon using a list of English opinion words, SentiWordNet, English–Urdu bilingual dictionary, and a collection of Urdu modifiers.

*Das & Bandyopadhyay (2010b)* proposed a computational method for generating an equivalent lexicon of English SentiWordNet using the English-Bengali bilingual dictionary. Their approach used a word-level translation process, which is followed by the error reduction technique. From the SentiWordNet, they selected a subset of opinion words whose orientation strength is above the heuristically identified threshold of 0.4. They used two Bengali corpora, News, and Blog to show the coverage of their developed lexicon. *Amin et al. (2019)*, the authors compiled a Bengali polarity lexicon from the English VADER lexicon using a translation technique. They modified the functionalities of the English VADER lexicon to make it fit for Bengali sentiment analysis.

## Comparison with existing sentiment lexicons

We provide comparisons with the existing Bengali lexicon-based methods in both the methodological level and evaluation phase. Due to the language difference, it is not possible to compare the proposed framework with the English lexicon-based methods in the evaluation step. To show the novelty and originality of the proposed framework, this section discusses how the proposed framework is different from the existing English sentiment lexicon creation methods at the methodological level.

### Bengali sentiment lexicons

In contrast to the existing Bengali sentiment lexicons, which are the simple word-level translation of English lexicons, BengSentiLex is created from a Bengali review corpus. Besides, BengSentiLex differs in the way it has been created. We use a cross-lingual corpus-based approach utilizing labeled and unlabeled data, while the existing lexicons merely translate the English sentiment lexicons to Bengali at the word level. Moreover, due to leveraging social media review data, BengSentilex contains opinion words people use in informal communication.

### English sentiment lexicons

This section discusses the key differences between the proposed framework and some of the English lexicon creation methods. A number of English lexicon creation methods employed label propagation algorithms utilizing provided seed words (*Velikovich et al., 2010*; *Hamilton et al., 2016*; *Tai & Kao, 2013*), Unlike them, BengSentiLex does not use any manual seed word list; rather, BengSentiLex finds a list of seed words from the corpus.

Some of the existing works integrated PMI or modified PMI in their lexicon generation framework (*Yang et al., 2013*; *Turney & Littman, 2003*; *Xu, Peng & Cheng, 2012*). Although PMI is employed to create 210 BengSentiLex, other than using it, the entire framework of BengSentiLex is different than the others methods. Besides, the above-mentioned methods calculate PMI among various features, while in BengSentiLex, PMI is employed between the feature and class-label.

The work of *Beigi & Moattar (2021)* utilized the frequency o words in positive and negative comments and vocabulary size of the corpus to determine the polarity score of the

corresponding word; in contrast, BengSentiLex uses PMI based sentiment intensity (SI) score to determine the semantic orientation of a word. Some other works utilized Matrix Factorization (*Peng & Park, 2011*) or distant supervision for creating lexicon (*Severyn & Moschitti, 2015*) for creating lexicon. A comprehensive literature review of the corpus-based lexicon creation method in English has been provided by *Darwich et al. (2019)*.

## STUDY RELATED TO PROFANITY AND ABUSIVE CONTENT ANALYSIS

Researchers studied the existence and sociolinguistic characteristics of swearing or cursing in social media. *Wang et al. (2014)* investigated the ubiquity, utility, and contextual dependency of swearing on Twitter. *Gauthier et al. (2015)* analyzed several sociolinguistic aspects of swearing on Twitter text data. Some studies investigated the relationship between gender and profanity (*Wang et al., 2014*; *Selnow, 1985*). The analysis revealed that males employ profanity much more often than females. Other social factors such as age, religiosity, or social status were also found to be related to the frequency of using vulgar words (*McEnery, 2004*). *Jay & Janschewitz (2008)* noticed that the offensiveness of taboo words depends on their contexts; taboo words used in conversational context are less offensive than the hostile context. *Pinker (2007)* classified the use of swear words into five categories: dysphemistic, abusive, idiomatic, emphatic, and cathartic.

Research related to the identification of swearing or offensive words has been predominantly conducted in English; Therefore, various lexicons comprised of profane or abusive words are available in English. *Pamungkas, Basile & Patti (2020)* created SWAD (Swear Words Abusiveness Dataset), a Twitter English corpus, where abusive swearing was manually annotated at the word level. Their collection consists of 1,511 unique swear words from 1,320 tweets. *Razavi et al. (2010)* manually collected approximately 2,700 dictionary entries including phrases and multi-word expressions. The lexicon creation process focusing the hate speech identification was reported by *Gitari et al. (2015)*. The largest English lexicon of abusive words was provided by *Wiegand et al. (2018)*.

In Bengali, several works investigated the presence of abusive language in social media data by leveraging supervised ML classifiers and labeled data (*Ishmam & Sharmin, 2019*; *Banik & Rahman, 2019*). *Emon et al. (2019)* utilized linear support vector classifier (LinearSVC), logistic regression(LR), multinomial naïve Bayes (MNB), random forest (RF), artificial neural network (ANN), recurrent neural network (RNN) with long short term memory (LSTM) to detect multi-type abusive Bengali text. They found RNN outperformed other classifiers by achieving the highest accuracy of 82.20%.

*Chakraborty & Seddiqui (2019)* employed various machine learning and natural language processing techniques to build an automatic system for detecting abusive comments in Bengali. As input, they used Unicode emoticons and Unicode Bengali characters. They applied MNB, SVM, Convolutional Neural Network (CNN) with LSTM to a dataset; they found SVM performed best with 78% accuracy.

*Karim et al. (2020)* proposed BengFastText, a word embedding model for Bengali, and incorporated it into a Multichannel Convolutional-LSTM (MConv-LSTM) network

for predicting different types of hate speech. They compared BengFastText against the Word2Vec and GloVe embedding by integrating them into several ML classifiers.

*Sazzed (2021a)* introduced an annotated Bengali corpus of 3000 transliterated Bengali comments categorized into two classes, abusive and non-abusive, 1500 comments for each. For the baseline evaluations, the author employed several supervised machine learning (ML) and deep learning-based classifiers. They observed support vector machine (SVM) shows the highest efficacy for identifying abusive content.

However, none of the existing works exclusively focused on creating resources to detect vulgarity or profanity in Bengali social media content. To the best of our knowledge, it is the first attempt to create a lexicon to detect vulgarity or profanity in the context of Bengali social media data.

# CREATION OF SENTIMENT LEXICON

## Basic terminologies

This section describes some of the concepts used in this paper for creating the sentiment lexicon.

### Supervisory characteristics

*Supervised learning.* Supervised learning is a popular way to train a machine learning model by utilizing annotated data. Using annotated inputs and outputs, the model can assess its accuracy and learn over time.

*Semi-supervised learning.* Semi-supervised learning uses both labeled and unlabeled data. It is a pragmatic approach when a high volume of data is available, but the annotation process is very challenging and requires a huge amount of time and resources.

### Cross-lingual approach

The cross-lingual approach leverages resources and tools from a resource-rich language (*e.g.*, English) to a resource-scarce language. Most of the existing study in sentiment analysis has been performed in English. Hence, resources from English can be employed in other languages using various language mapping techniques. The construction of a language-specific sentiment lexicon requires vast resources, tools, and an active research community, which are not available in the resource-scarce language. A feasible approach could be utilizing resources from the languages where sentiment resources are abundant (*Sazzed, 2021b*). In this work, we employ machine translation to leverage several resources from English.

### Machine translation

Machine translation (MT) refers to the use of software to translate text or speech from one language to another. Over the decades, the machine translation system has evolved to a more reliable system, from the simple word-level substitution to sophisticated Neural Machine Translation (NMT) (*Kalchbrenner & Blunsom, 2013*; *Bahdanau, Cho & Bengio, 2014*; *Zhu et al., 2020*). Machine translation has been successfully applied to various sentiment analysis tasks

| Bengali Reviews | Machine Translation | Polarity |
|---|---|---|
| শামিম ভাইয়ের কাছে এমন নাটক আশা করি নি! ! | I did not expect such a drama from Shamim Bhai !! | Negative |
| ফালতু নাটক কবির সিং মুভির ট্রেইলার কপি করে সাওয়ার নাটক বানায়। | False drama Kabir Singh copied movie trailer and made Sawar drama. | Negative |
| যখন মন খুব থারাপ থাকে,তখন আপনাদের নাটক দেখি।তখন মনটা আরো থারাপ হয়ে যায়,আর একটা সময়ে মন থারাপে, মন থারাপে কাটাকাটি।সত্যি অনেক ভাল লাগে আপনাদের অভিনয় গুলা। দোয়া রইলো চালিয়ে যান ভাই।।।।। | When the mind is very bad, then I watch your dramas. The prayer continued, brother | Positive |
| আফরান নিশো ভাই আমাদের টাংগাইলের অহংকার | Afran Nisho Bhai is the pride of our Tangail | Positive |
| অসাধারণ একটা জুটির অসাধারণ একটা নাটক ছিল ।।।। শেষ ৫ মিনিট ভীষণ কষ্ট লাগলো।।।। বারবার দেখতে ইচ্ছে করছে ।।।। | An extraordinary pair had an extraordinary drama … The last 5 minutes were very difficult … Wanting to see again and again … | Positive |
| পরিচালক একটা গাঁজা খোর ছাগলের বাচ্চা এইসব কী | The director is a cannabis-eating goat kid | Negative |
| অসাধারন আমার কাছে সেই লাকছে~ | Extraordinary That Looks To Me | Positive |

**Figure 1** Sample reviews from training dataset Drama-Train.

by researchers. *Balahur & Turchi (2014)* studied the possibility of employing machine translation systems and supervised methods to build models that can detect and classify sentiment in low-resource languages. Their evaluation showed that machine translation systems were rapidly maturing. The authors claimed that with appropriate ML algorithms and carefully chosen features, machine translation could be used to build sentiment analysis systems in resource-poor languages.

### Pointwise mutual information (PMI)

Pointwise Mutual Information (PMI) is a measure of association used in information theory and statistics. The PMI between two variables X and Y is computed as,

$$PMI(X, Y) = \log \frac{P(X, Y)}{P(X)P(Y)} \tag{1}$$

The term P(X, Y) depicts the number of observations of the co-occurrences of event X and Y. P(X) represents the number of times X occurs, and P(Y) means the number of times Y occurs. When two variables X and Y are independent, the PMI between them is 0. PMI maximizes when X and Y are perfectly correlated.

## Datasets for lexicon creation and evaluation
### Training dataset for sentiment lexicon

We use a drama review dataset (*Drama-Train*) (*Sazzed, 2020a*) collected from YouTube to build the BengSentiLex. This review corpus consists of around 50,000 Bengali reviews, where each review represents the viewer's opinions towards a Bengali drama. Among the 50,000 Bengali reviews, around 11,807 were annotated by *Sazzed (2020a)*, while the remaining reviews are unlabeled. Figure 1 shows examples of drama reviews belong to the *Drama-Train*.

**Table 1  Evaluation datasets for BengSentiLex.**

| Dataset | Domain | Positive | Negative | Total |
|---|---|---|---|---|
| **Drama-Eval** | Drama Review | 1000 | 1000 | 2000 |
| **News1** | News Comments | 2000 | 2000 | 4000 |
| **News2** | News Comments | 5205 | 5205 | 10410 |

### Evaluation dataset for sentiment lexicon

We show the effectiveness of BengSentiLex in three datasets from distinct domains. Table 1 provides the details of the evaluation datasets.

The first evaluation dataset is a drama review dataset (*Drama-Eval*) that comprises 1000 annotated reviews. This dataset belongs to the same domain as the training dataset, *Drama-Train*; However, it was not used for the lexicon creation. This is a class-balanced dataset, consists of 500 *positive* and 500 *negative* reviews.

The second dataset is a news dataset, *News1*, consists of 4,000 news comments (https://github.com/aimansnigdha/Bangla-Abusive-Comment-651Dataset.git); among them, 2000 are *positive* and 2000 are *negative* comments.

The third dataset is also a news comment dataset, (*News2*), collected from two popular Bengali newspapers, Prothom Alo and BBC Bangla (*Taher, Akhter & Hasan, 2018*). It consists of 5205 positive comments and 5600 negative comments. For the evaluation, we select a class-balanced subset where each class contains 5205 comments.

## Cross-lingual resources
### Sentiment lexicon

To identify opinions conveying Bengali words, we employ a cross-lingual approach. By leveraging machine translation and English sentiment lexicons, we determine whether an extracted Bengali word expresses any sentiment. The study by *Sazzed & Jayarathna (2019a)* showed that though the Bengali to English machine translation (*i.e.,* Google Translate) system is not perfect, it preserves semantic orientation in most cases.

We translate all the extracted Bengali words into English and find their polarities based on the English sentiment lexicons. The occurrence of a translated word in an English opinion lexicon indicates that it conveys opinion; hence, the corresponding Bengali word can be included in BengSentiLex. Although we perform word-level translation between Bengali and English, it is different from the existing works as words are translated from Bengali to English, not the other way. By translating words from Bengali to English, the proposed framework can capture words used in informal or social communication, which is not possible to achieve when dictionary-based translations of the English lexicons are used.

Furthermore, this approach can yield and include multiple synonymous opinion words instead of one. For example, by translating an English sentiment word to Bengali provides only the corresponding Bengali word. However, when corpus extracted words are translated to English, due to the low coverage of the machine translation system, synonymous Bengali words are often be mapped into the same English polarity word. This actually helps to identify and include more opinion words in the Bengali lexicon.

To determine the polarity of the translated words, we utilize the following English sentiment lexicons, opinion lexicon, and VADER. Opinion lexicon was developed by (*Hu & Liu, 2004*) and contains around 6800 English sentiment words (*positive* or *negative*). Besides the dictionary words, it also includes acronyms, misspelled words, and abbreviations. The opinion lexicon is a binary lexicon, where each word is associated with either *positive* (+1) or *negative* (−1) polarity score. VADER is a sentiment lexicon especially attuned to social media text. VADER contains over 7,500 lexical features with sentiment polarity of either *positive* or *negative* and sentiment intensity score between –4 to +4.

### Part-of-speech (POS) tagging

A Part-of-speech (POS) tagger is a tool that assigns a POS tag (*e.g.*, noun, verb, adjective, *etc.*) to each of the words present in a text. As adjectives, nouns, and verbs usually convey opinions, the POS tagger can help the presence of opinion words. In English, several standard POS taggers are available such as NLTK POS tagger *Loper & Bird (2002)* and spaCy POS tagger *Honnibal & Montani (2017)*. In Bengali, due to the lack of any sophisticate POS tagger is available, we use the machine translation system to convert the likely Bengali opinion words to English. We then use the spaCy POS tagger to determine the POS tag of those English words, which allows us to label the POS tag of the corresponding Bengali words.

## Methodology

The creation of the BengSentiLex involves several steps. We utilize various tools and resources to find opinion conveying words from the corpus and include them to BengSentiLex, as shown in Fig. 2.

- Phase 1: Labeled and unlabeled corpus, machine translation system, English lexicons.
- Phase 2: Labeled corpus, PMI, machine translation system, English POS tagger, English lexicons, Bengali lexicon constructed in phase 1.
- Phase 3: Unlabeled corpus, ML classifiers, PMI, machine translation system, English POS tagger, English lexicons, Bengali lexicon (constructed in phase 1 and phase 2).

Each phase expands BengSentiLex with the newly recognized opinion-conveying words. A manual validation step is included to examine the identified opinion words. Afterward, we generate synonyms for the validated words, which are added to the lexicon.

For synonym generation, we utilize Google Translate (https://translate.google.com), as no standard Bengali synonym dictionary is available on the web in digital format. We translate Bengali words into multiple languages such as English, Chinese, French, Hindi, Russian and Arabic and then perform back-translation. This approach assists in finding synonyms as sentiments are expressed in different ways across the languages.

### Phase-1: Utilizing English sentiment lexicons

The development of a sentiment lexicon typically starts with a list of well-defined sentiment words. A well-known approach for identifying the initial list of words (often called seed words) is to use a dictionary. However, dictionary words denote mainly formal expressions and usually do not represent the words people use in social media or informal

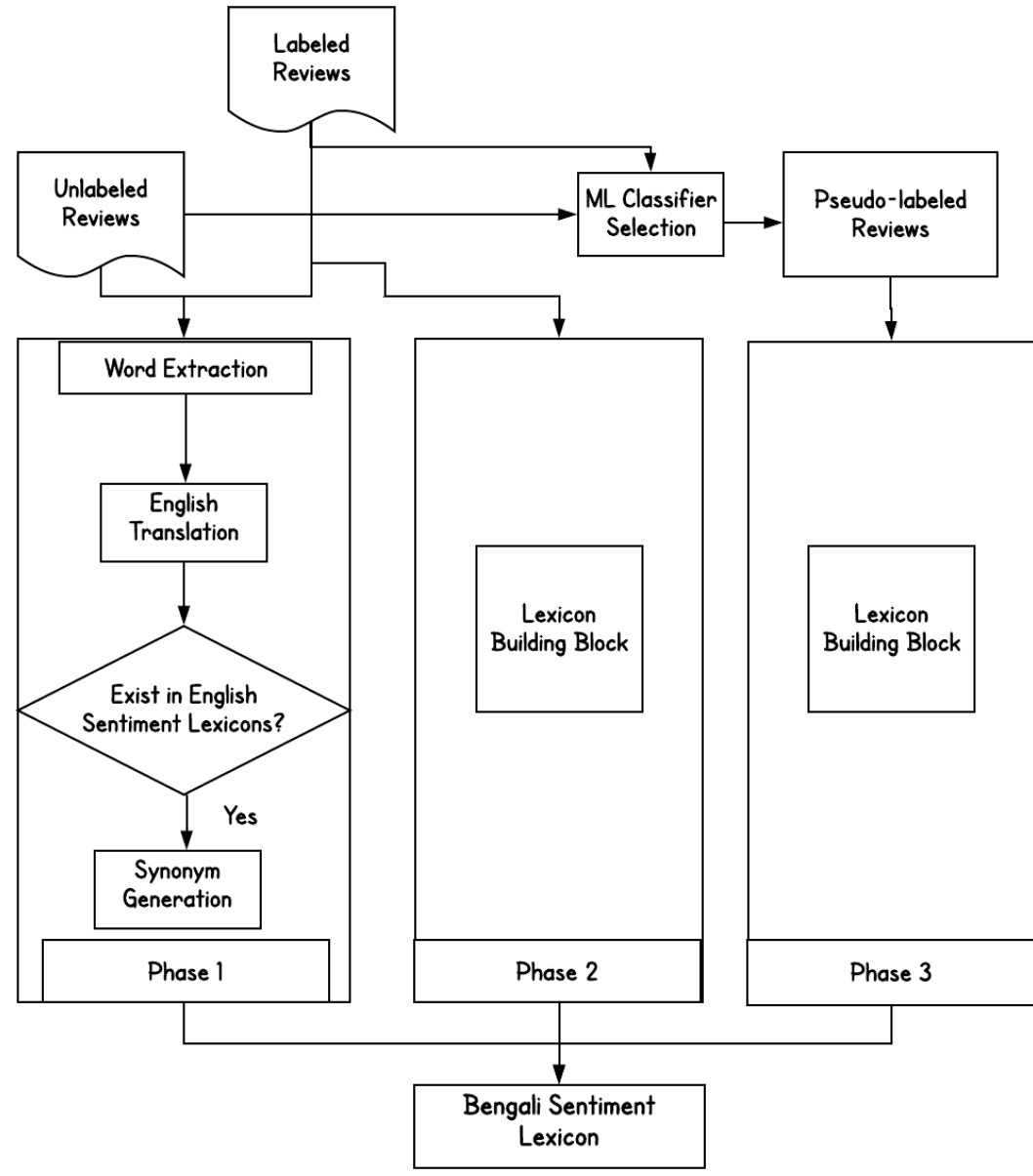

**Figure 2** The various phases of sentiment lexicon generation in Bengali.

communication. On the contrary, words extracted from a corpus represent expressions people use in regular communication; hence, more useful for sentiment analysis.

We tokenize words from the review corpus (both labeled and unlabeled) using the NLTK tokenizer and calculate their frequency in the corpus. The words with a frequency above five are added to the candidate pool. However, not all high-frequency words convey sentiments. For example-'drama' is a high-frequency word in the drama review dataset, but it is not a sentiment word. As Bengali does not have any sentiment lexicon of its own, we utilize resources from English. Using a machine translation system, we translate all the

words from the candidate pool to English. Two English sentiment lexicons, opinion lexicon, and VADER are employed to determine the polarity of the translated words. If a translated English word exists in the English sentiment lexicon, then it is an opinion conveying word; therefore, the corresponding Bengali word is added to the Bengali sentiment dictionary.

### Phase 2: Lexicon generation from labeled data

Phase 2 retrieves opinion words from the labeled corpus by leveraging the pointwise mutual information (PMI) and a POS tagger, as shown in Figs. 2 and 3.

From the labeled reviews, we derive the words that are highly correlated with the class label. The words or terms that already exist in the lexicon (identified in earlier phases) are not considered. The remaining words are translated into English using the machine translation system. We utilize the spaCy POS tagger to identify their POS tags. Since usually adjectives and verbs convey opinions, we only keep them and exclude the other POS.

The sentiment score of a word, $w$, is calculated using the formula shown below,

$$SentimentScore(w) = PMI(w, pos) - PMI(w, neg) \tag{2}$$

where, $PMI(w, pos)$ represents the PMI score of word $w$ corresponding to *positive* class and $PMI(w, neg)$ represents the PMI score of word $w$ corresponding to *negative* class.

We then calculate the sentiment intensity (SI) of $w$, using the following equation,

$$SI(w) = \frac{SentimentScore(w)}{PMI(w, pos) + PMI(w, neg)} \tag{3}$$

We compare the sentiment strengths of words with the threshold value to identity opinion conveying words from the labeled reviews.

If the sentiment intensity of a word, $w$, is above the threshold of 0.5, we consider it as a *positive word*. if sentiment strength is below -0.5, we consider it as a *negative* word.

$$Class(w) = \begin{cases} Positive, & if \ SI(w) > 0.50 \\ Negative, & if \ SI(w) < -0.50 \\ Unassigned, & Otherwise \end{cases} \tag{4}$$

### Phase 3: Lexicon generation from unlabeled (pseudo-labeled) data

In addition to annotated reviews, the *Drama-Train* corpus consists of a large number of unlabeled reviews. For the labeled reviews, we use PMI to identify the top class-correlated words. However, for the unannotated reviews, no class labels are available; thus, automatic labeling is required. To automatically annotate the unlabeled reviews, we employ various several ML classifiers and select the one with the highest accuracy. The unigram and bigram-based tf–idf (term frequency-inverse document frequency) scores are used as input features for the ML classifiers. The following ML classifiers are employed.

Support Vector Machine (SVM) is a popular supervised ML algorithm used for classification and regression problems. Originally, SVM is a binary classifier that decides the best hyperplane to separate the space into binary classes by maximizing the distance between data points belong to different classes. However, SVM can be used as multi-class classifier following same principle and employing *one-versus-one* or *one-vs-the-rest* strategy.

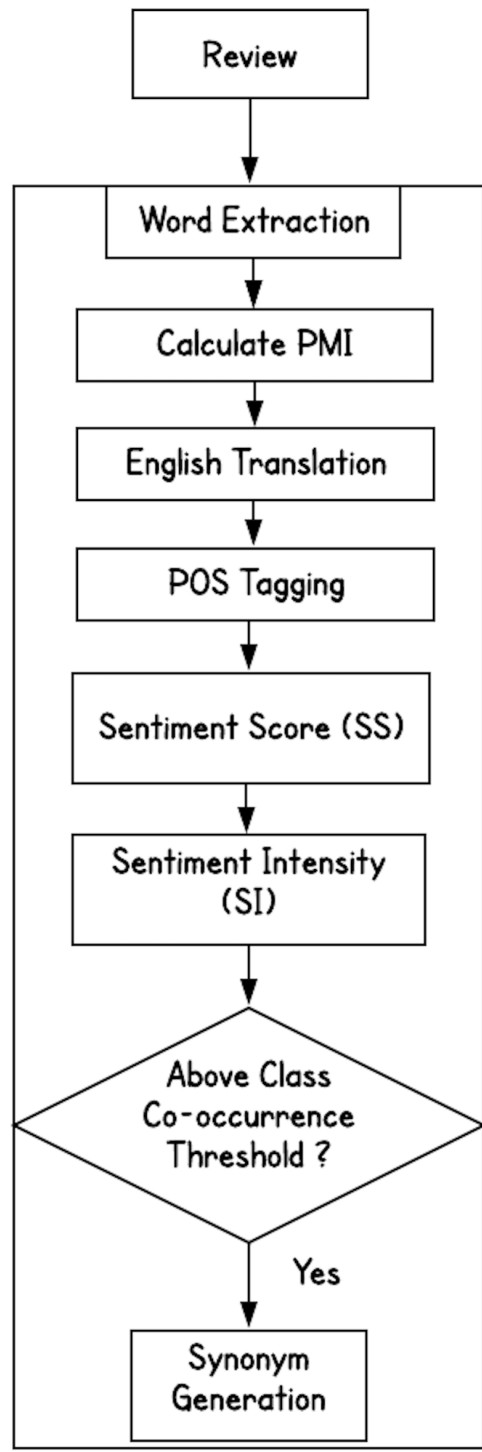

**Figure 3** The lexicon building block.

Stochastic Gradient Descent (SGD) is a optimization method that optimizes an objective function iteratively. It is a stochastic approximation of actual gradient descent optimization

**Table 2  Performances of supervised ML classifiers in annotated corpus.**

| Classifier | Precision | Recall | F1 Score | Accuracy |
| --- | --- | --- | --- | --- |
| SGD | 0.939 | 0.901 | 0.920 | 93.61% |
| SVM | 0.908 | 0.924 | 0.916 | 93.00% |
| LR | 0.889 | 0.922 | 0.905 | 91.80% |
| k-NN | 0.901 | 0.849 | 0.875 | 90.18% |
| RF | 0.878 | 0.870 | 0.874 | 89.9l% |

since it calculates gradient from a randomly selected subset of the data. For SGD, we use hinge loss and l2 penalty with a maximum iteration of 1,500.

Logistic regression (LR) is a statistical classification method that finds the best fitting model to describe the relationship between the dependent variable and a set of independent variables.

Random Forest (RF) is a decision tree-based ensemble learning classifier. It makes predictions by combining the results from multiple individual decision trees.

K-nearest neighbors (k-NN) algorithm is a non-parametric method used for classification and regression. In k-NN classification, the class membership of a sample is determined by the majority voting of neighbors. Here, we set the value of k to 3, which means the class of a review depends on three of its closest neighbors.

We use scikit-learn (*Pedregosa et al., 2011*) implementation of the aforementioned ML classifiers. For all of the classifiers, we use the default parameter settings with class-balanced weight.. Using 10-fold cross-validation, we assess their performances. The purpose of this step is to find reliable classifiers that can be used for automatic class labeling.

Table 2 shows the classification accuracy of various ML classifiers based on 10-fold cross-validation. Among the five classifiers we employ, SGD and SVM show higher accuracy. Both of them correctly identify around 93% of the reviews, which is close to the accuracy of manual annotations. LR shows similar accuracy of around 92%. We use these three classifiers to determine the class of the unlabeled reviews.

The following two approaches are considered for automatically generating the class label of the unannotated reviews utilizing the ML classifiers,

*Approach-1*: Using the labeled reviews as training data and all the unlabeled reviews as testing data.

*Approach-2*: Iteratively utilizing a small unlabeled set as testing data. After assigning their labels, adding these pseudo-labeled reviews to the training set, and selecting a new set of unlabeled reviews as testing data. This procedure continues until all the data are labeled.

To determine the performance of the *approach-1*, we conduct 4-fold cross-validation on the labeled reviews. We use 1-fold as training data and the remaining 3-folds as testing data. The training and testing dataset ratio reflects the ratio of labeled reviews (11807) and unlabeled reviews (nearly 38000). For approach-2, similar way to approach-1, we split the 11807 labeled reviews into four subsets and use one subset (about 3000 reviews) as a training set. However, the testing set selection process differs. In each iteration, a testing set is randomly selected from the remaining three subsets (around 9000 reviews). The

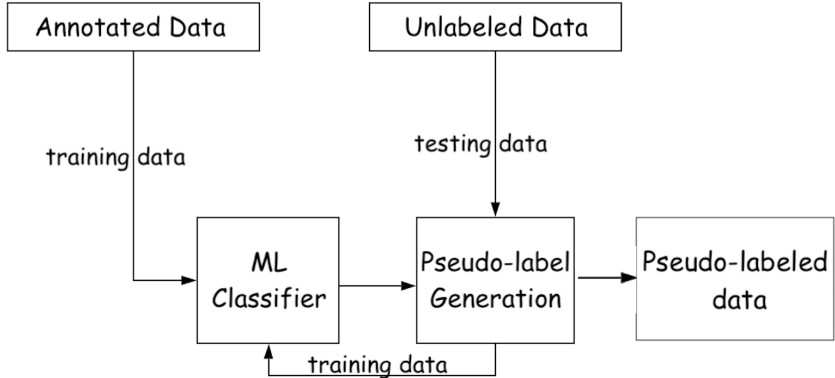

Steps for generating pseudo-labeled data

**Figure 4** Class-label assignment of unlabeled reviews using supervised ML classifier and labeled data.

testing set size is made equal to at most 10% of the current training set. After assigning the predictions for the reviews of the current testing set, they are added to the training set, and a new testing set is selected following the same criteria. This process continues until all the reviews of three subsets (about 9000 reviews each) are annotated.

We find that gradually expanding the training set by adding the predicted reviews from the testing set provides better performance (*i.e., approach-2*). After applying *i.e., approach-2* (shown in Fig. 4), the dataset contains around 38,000 pseudo-labeled reviews, classified by the majority voting of SGD, SVM, and LR. We then employ PMI and POS tagger in a similar way to phase 2. However, since this phase utilizes pseudo-labeled data instead of the true-label data, we set a higher threshold of 0.7 for the class label assignment.

# CREATION OF SWEAR LEXICON
## Corpus
We utilize two Bengali corpora, one for creating the swear lexicon, BengSwearLex, which we refer to as *training corpus (SW)*, and the other one for analysis and evaluating the performance of BengSwearLex, which we refer to as *evaluation corpus (SW)*.

### Training corpus (SW)
The training corpus (SW) consists of 10221 Bengali comments that belong to different categories, such as toxic, racism, obscene, and insult (https://translate.google.com). We notice the presence of many empty and punctuation-only comments and erroneous annotations in the corpus. We exclude all the comments having the above-mentioned issues. From the modified corpus, we only select the reviews labeled as *obscene*. After excluding erroneous reviews and reviews that belong to other classes, the final corpus consists of 3902 obscene comments. The length of each comment ranges from 1-100 words.

| Bengali Comments | English Translations |
|---|---|
| বাংলা নাটকের গোয়া মোশাররফ করিম গং রাই মারতাছে, | Mosharraf Karims gang's are fucking Bengali drama, |
| চুদনাগিরি স্ক্রিপ্ট ছাড়া আর কোন স্ক্রিপ্ট ছিলো না।মাদারচোদ মার্কা নাটক এইটা | Wasn't there any other script except this fucking one. This is a motherfucker drama. |
| ব্লাইঙ্খোদ থানকির ছেলে। এতো অ্যাড চুদাও কে,,, | Fucker whore's son. why so many advertisements? |

**Figure 5** Examples of Bengali obscene comments and corresponding English translation.

**Table 3** Description of drama review evaluation corpus.

| Profane | Non-Profane | Total |
|---|---|---|
| 664 | 2643 | 3307 |

Figure 5 shows some examples of the obscene comments from the *training corpus (SW)*.

### Evaluation corpus (SW)

The *evaluation corpus (SW)* we utilize is a drama review corpus collected from YouTube. This corpus was created and deposited by (*Sazzed, 2020a*) for sentiment analysis; It consists of 8,500 positive and 3,307 negative reviews. However, there is no distinction between different types of negative reviews. Therefore, we manually label these 3,307 negative reviews into two categories, profane and non-profane. The annotation of these 3307 negative reviews was performed by three Bengali native speakers (A1, A2, A3). The first two annotators (A1 and A2) initially annotated all the reviews. In case of disagreement in annotation, it was resolved by the third annotator (A3).

After annotation, this corpus consists of 2643 non-profane negative reviews and 664 profane reviews, as shown in Table 3. The kappa ($k$) statistic shows a value of 0.81 for the two raters (A1 and A2) is 0.81, which indicates a high agreement in the annotation.

## Text processing tools
### POS tagger

Similar to sentiment lexicon creation framework, here we utilize a POS tagger to identify opinion word (https://github.com/AbuKaisar24/Bengali-Pos-Tagger-Using-Indian-Corpus/ https://www.isical.ac.in/ utpal/docs/POSreadme.txt). Since the existing Bengali POS taggers are not as accurate as of its English counterpart, manual validation is needed to check the correctness of the POS tags assigned by the POS taggers.

### Word embedding

A word embedding is a learned representation for text, where related words have similar representations. The word embedding technique provides an efficient way to use the dense representation of words of varying lengths. The word embedding values are learned by the model during the training phase.

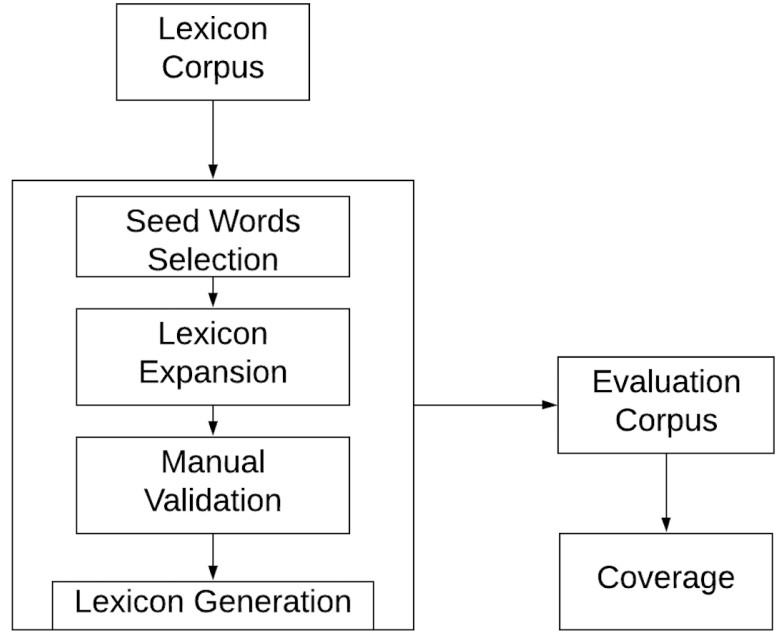

**Figure 6** **The proposed methodology.**

There exist two main approaches for learning word embedding, count-based and context-based. The count-based vector space models heavily rely on the word frequency and co-occurrence matrix with the assumption that words in the same contexts share similar or related semantic meanings. The other learning approach, context-based methods, build predictive models that predict the target word given its neighbors. The best vector representation of each word is learned during the model training process. The Continuous Bag-of-Words (CBOW) model is a popular context-based method for learning word vectors. It predicts the center word from surrounding context words.

## Swear Lexicon creation framework

The proposed semi-automatic approach utilizes annotated corpus, word embedding, and POS tagger. The entire framework consists of three phases as shown in Fig. 6,

1. Seed word selection
2. Lexicon expansion
3. Manual validation

### Seed word selection

The lexicon creation process usually begins with a list of seed words. The proposed approach utilizes an annotated obscene corpus to generate a seed word list. We extract and count the occurrence of individual words in the corpus. Based on the word-occurrence count in the corpus, we select the top 100 words. However, we find that the list contains some non-vulgar words, which we exclude utilizing POS tagger and manual validation.

### Lexicon expansion

The lexicon expansion step involves utilizing word embedding to identify similar words of the already recognized swear words. We use the training corpus (SW) as a training dataset and utilize the Gensim (*Rehurek & Sojka, 2010*) Continuous Bag-of-Words (CBOW) implementation to find similar words.

The entire procedure consists of the following steps-

- In the first step, we identify words that are most similar to the selected seed words.
- The second step iteratively recognizes words in the corpus which are similar to obscene words of BengSwearLex. We remove the duplicate word automatically. In addition, we remove words that are not a noun, adjective, or verb. Several iterations are performed until we notice no significant expansion of the swear word list.

### Manual validation

In the final step, we manually exclude non-swear words that exist in the swear lexicon. As lexical resources such as POS tagger in Bengali are not sophisticated enough, a manual validation step is necessary to eliminate non-swear words. Moreover, we find that profane comments often do not follow the usual sentence structure; Therefore, the POS tagger fails to label them correctly.

## EXPERIMENTAL RESULTS

### Sentiment classification

#### Baselines and evaluation metrics

We compare the performances of the corpus-built lexicon BengSentiLex (716 negative words and 519 positive words) with the Bengali translation of three English sentiment lexicons: VADER (7518 words), AFINN (2477 words), and Opinion Lexicon (6800 words) by integrating them into a lexicon-based classifier. We compute the accuracy of all these four lexicons in three cross-domain evaluation datasets to demonstrate the efficacy of BengSentiLex for sentiment classification in varied domains and distributions.

As no lexicon-based sentiment analysis tool is publicly available in Bengali, we develop a simple lexicon-based sentiment analysis method, BengSentiAn, that integrates a sentiment lexicon for predicting the class label of a review. The polarity score of a review is computed by summing up the polarity score of individual opinion words provided by the sentiment lexicon. Besides, negation words are considered address the shift of polarity of the opinion words. A polarity score above 0 for a review is considered as a positive prediction, while a prediction with a polarity score below 0 indicates a negative prediction. When a 0 polarity score is obtained using a lexicon, we consider the prediction as wrong. A polarity score of 0 can be obtained when the word-level polarity score of a lexicon can not distinguish a review as positive or negative, or the lexicon lacks coverage of opinion words present in the review. It is more appropriate to consider this scenario as a misprediction rather than considering a positive or negative class prediction.

**Table 4  Accuracy of various lexicons for sentiment classification.**

| Dataset | Lexicon | #Neg Class | #Pos Class | Total |
|---|---|---|---|---|
| Drama | AFINN | 145/1000 (14.15%) | 488/1000 (48.80%) | 633/2000 (31.65%) |
| | Opinion Lexicon | 241/1000 (24.10%) | 598/1000 (59.80%) | 839/2000 (41.95%) |
| | VADER | 225/1000 (22.5%) | 707/1000 (70.70%) | 932/2000 (46.60%) |
| | BengSentiLex | 533/1000 (53.30%) | 775/1000 (77.50%) | 1308/2000 (65.40%) |
| News1 | AFINN | 626/2000 (31.3%) | 628/2000 (31.4%) | 1254/4000 (31.35%) |
| | Opinion Lexicon | 590/2000 (29.5%) | 833/2000 (41.65%) | 1423/4000 (35.57%) |
| | VADER | 566/2000 (28.30%) | 1070/2000 (53.50%) | 1636/4000 (40.90%) |
| | BengSentiLex | 932/2000 (46.6%) | 960/2000 (48.80%) | 1892 4000 (47.30%) |
| News2 | AFINN | 2004/5660 (35.4%) | 1826/5205 (35.08%) | 3830/10865 (35.2%) |
| | Opinion Lexicon | 1763/5660 (31.14%) | 2274/5205 (43.68%) | 4037/10865 (37.15%) |
| | VADER | 1662/5205 (31.93%) | 2827/5205 (54.31%) | 4489/10410 (43.12%) |
| | BengSentiLex | 2086/5205 (40.08%) | 2721/5205 (52.27%) | 4807/10410 (46.17%) |

## Comparative results

Table 4 shows the comparative performances of several translated lexicons and BengSentiLex after integrating them into BengSentiAn. In the drama review dataset, BengSentiLex classifies 1308 reviews out of 2000 reviews with an accuracy of around 65%. Among the three translated lexicons, VADER shows an accuracy of 46.60%; AFINN and opinion lexicon provide 31.65% and 41.95% accuracy, respectively. In the News1 dataset, BengSentiLex exhibits an accuracy of 47.30%, while the VADER, opinion lexicon, and AFINN provide an accuracy of 40.90%, 35.57%, 31.65%, respectively. In the News2 dataset, BengSentiLex shows an accuracy of 46.17%, while the VADER provides the second-best performance with an accuracy of 43.12%.

## Profanity identification
### Evaluation metric

To show the effectiveness of BengSwearLex, we utilize document-level coverage, which is similar to the recall score ($R_{profane}$) of the ML classifiers for profane class identification. The document-level coverage (or recall) of a lexicon corresponding to a review corpus is calculated as follows-

From the corpus, we first count the number of reviews that contain at least one word from the lexicon, which is then divided by the total number of reviews present in the corpus. Finally, it is multiplied by 100. The following equation is used to calculate document-level coverage ($DCov$) of a lexicon-

$$DCov = \frac{Number\ of\ reviews\ containing\ at\ least\ one\ swear\ word\ from\ the\ lexicon}{total\ number\ of\ reviews\ in\ corpus} * 100$$

As BengSwearLex tries to identify text that contains swearing, vulgar, obscene, or slang words, the document-level coverage is provided only for the profane or vulgar reviews. Besides, BengSwearLex is manually validated at the final step to ensure that it contains only filthy words; hence, it is unlikely that it recognizes a non-profane comment as profane (false

positive). As no swear or obscene lexicon exists in Bengali, we compare the performance of BengSwearLex with several supervised classifiers (that use in-domain labeled data) for profanity detection in the evaluation corpus.

As no swear lexicon exists in Bengali, we compare the performance of BengSwearLex with several supervised classifiers (that use in-domain labeled data) for profanity detection in the evaluation corpus.

Two classical ML classifiers, Logistics Regression (LR) and Support Vector Machine (SVM), and an optimization method, Stochastic Gradient Descendent (SGD,) are employed in the evaluation corpus to identify profane reviews. As the input features, we use unigram and bigram-based term frequency-inverse document frequency(tf-idf) scores. 10-fold cross-validation is performed to assess the performance of various ML classifiers. For all the classifiers, default parameter settings are used. For SGD, the hinge loss with l2 penalty is used.

Furthermore, we employ Deep Neural Network (DNN) based architecture, Convolutional Neural Network (CNN), Long short-term memory (LSTM), and Bidirectional Long short-term memory (BiLSTM) to identify profanity. The DNN based model starts with the Keras (*Chollet et al., 2015*) embedding layer. The three important parameters of the embedding layer are *input dimension*, which represents the size of the vocabulary, *output dimensions*, which is the length of the vector for each word, *input length*, the maximum length of a sequence. The *input dimension* is determined by the number of words present ina corpus, which varies in two corpora. We set the *output dimensions* to 64. The maximum length of a sequence is used as 300. A drop-out rate of 0.5 is employed to the dropout layer; ReLU activation is used in the intermediate layers. In the final layer, softmax activation is applied. As an optimization function, Adam optimizer, and as a loss function, binary-cross entropy are utilized. We set the batch size to 64, use a learning rate of 0.001, and train the model for 10 epochs. We use the Keras library (*Chollet et al., 2015*) with the TensorFlow backend for implementing DNN based model.

### Comparison results

Table 5 shows that among the 664 profane reviews present in the evaluation corpus (SW), BengSwearLex registers 564 reviews as profane by identifying the presence of at least one swear term in the review. BengSwearLex yields a DConv score of 84.93% in the evaluation corpus (SW).

Table also 5 provides the coverage of all the ML classifiers in the evaluation corpus. We provide their performances in two different settings: class-balanced setting and class-imbalanced setting. The original class-imbalanced setting contains all the 664 profane comments and 2643 non-profane negative comments. In the class-balanced setting, 664 non-profane comments are randomly selected from the list of 2643 non-profane comments to make the dataset class-balanced.

From Table 5, we observe that when the original class-imbalanced data is used, all the three ML classifiers achieve coverage of around 60%. However, in a class-balanced dataset is utilized, the performances of ML classifiers dramatically increase, reach to coverage of around 94%.

**Table 5** Document-level coverage of various methods for profanity detection.

| Type | Method | # Identified | DCov |
|---|---|---|---|
| Unsupervised | **BengSwearLex** | 564/664 | 84.93% |
| | **LR** | 161/664 | 24.5% |
| Supervised (Unbalanced) | **SVM** | 345/664 | 53.4% |
| | **SGD** | 366/664 | 58.8% |
| | **LSTM** | 433/664 | 65.21% |
| | **BiLSTM** | 462/664 | 70.4% |
| | **CNN** | 444/664 | 66.86% |
| | **LR** | 609/664 | 91.71% |
| Supervised (Balanced) | **SVM** | 594/664 | 89.45% |
| | **SGD** | 589/664 | 88.70% |
| | **LSTM** | 610/664 | 91.67% |
| | **BiLSTM** | 624/664 | 94.0% |
| | **CNN** | 609/664 | 91.71% |

## DISCUSSION

### Sentiment Lexicon

The results suggest that translated lexicons can not adequately capture the semantic orientation of the Bengali reviews as they lack coverage of opinion words presents in Bengali text. We find that BengSentiLex performs considerably better than the translated lexicons in the drama review dataset, with over 40% improvements. Since BengSentiLex is developed from the corpus that belongs to the same domain, it is very effective at classifying sentiments in the *Drama-Eval* corpus as it is created from the corpus of the same domain.

Also, for the two other cross-domain evaluation corpus, News1 and News2, BengSentiLex yields better performance compared to translated lexicons; especially, for classifying negative reviews, which can be attributed to the presence of a higher number of *negative* opinion words (716) in BengSentiLex compared to 519 *positive* sentiment words.

The results indicate that utilizing corpus in the target language for automated sentiment lexicon generation is more effective compared to translating words directly from another language such as English. As BengSentiLex is built from a social media corpus, it is comprised of words that people use on the web, social media, and informal communication; Therefore, it is more effective in recognizing sentiments in Bengali social media data compared to word-level translated lexicons.

Although supervised ML classifiers usually perform better in sentiment classification, they require annotated data that are mostly missing in low-resource languages such as Bengali. The developed lexicon can help to deal with the inadequacy of labeled review data in Bengali.

### Swear Lexicon

The results of Table 5 reveal that BengSwearLex is capable of identifying profanity in Bengali social media content. It shows higher document-level than ML classifiers when

a class-imbalanced training set is used. Nonetheless, ML classifiers leveraging a class-balanced training set performs better than BengSwearLex. Labeled data is scarcely available in low-resource languages such as Bengali; therefore, although small in size, BengSwearLex can be an effective tool for profanity or obscenity identification in the lack of labeled data. Besides, since BengSwearLex consists of only swear or obscene terms, there is a very low possibility that it would refer to non-obscene comments as obscene obscene (false positive); thus, BengSwearLex is capable of achieving a very high precision score for obscene review identification.

## SUMMARY AND CONCLUSIONS

In this paper, we present two methodologies for creating lexical resources for Bengali (*i.e.,* sentiment lexicon and swear/obscene lexicon). The first methodology leverages the Bengali review corpus, machine-translation system, English lexicons, and ML classifiers to develop the Bengali sentiment lexicon, BengSentiLex. We demonstrate the effectiveness of BengSentiLex in both in-domain and cross-domain datasets. When integrated into a lexicon-based tool, BengSentiLex yields better performance compared to translated English lexicons. The other methodology creates the swear lexicon, named BengSwearLex, utilizing an obscene corpus and other text processing tools and resources. We show that BengSwearLex is capable of identifying vulgar language that exists in social media content. Besides, we provide an annotated dataset for profanity analysis in Bengali social media data. We have made both BengSentiLex (https://github.com/sazzadcsedu/BNLexicon.git) and BengSwearLex (https://github.com/sazzadcsedu/Bangla-Vulgar-Lexicon.git) publicly available for the researchers.

The superior performance of the BengSentiLex suggests that the corpus-based lexicon can capture the language-specific features and connotations related to the language, which translated sentiment lexicons can not do. Similarly, BengSwearLex can be utilized to distinguish profanity in Bengali social media content when annotated data are unavailable. The proposed frameworks can be adopted in other resource-limited languages to create lexical resources. The future work will involve expanding the size of both lexicons utilizing larger and multi-domain training corpora.

### Funding
The authors received no funding for this work.

### Competing Interests
The authors declare there are no competing interests.

### Author Contributions
- Salim Sazzed conceived and designed the experiments, performed the experiments, analyzed the data, performed the computation work, prepared figures and/or tables, authored or reviewed drafts of the paper, and approved the final draft.

## Data Availability

The code and data used in this article are available at GitHub: https://github.com/sazzadcsedu/BNLexicon, https://github.com/sazzadcsedu/Bangla-Vulgar-Lexicon.

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
