# Peer review of "BengSentiLex and BengSwearLex: creating lexicons for sentiment analysis and profanity detection in low-resource Bengali language"

_PeerJ Computer Science, doi:10.7717/peerj-cs.681_

## Round 0.1 · original submission · Major Revisions

Kindly modify the paper as per the reviewers comments and suggestions.

Reviewer 1 ·

Basic reporting

no comment

Experimental design

The discussion of the experiment needs to be more in-depth, and the content of the experiment needs to be more detailed and complete. Please analyze the advantages of the proposed method and supplement the related experiments.

Validity of the findings

no comment

Additional comments

1. Your most important issue
Q1: Are the methods presented in this paper the most advanced and original, what are the main challenges of this work, and these need to be further discussed in the relevant work.
Q2: The discussion of the experiment needs to be more in-depth, and the content of the experiment needs to be more detailed and complete. Please analyze the advantages of the proposed method and supplement the related experiments.
2. The next most important item
Q3: 3.3.2 What is the meaning of the PMI formula and its variables? Please give specific explanation.
Q4: 3.3.3 “After applying approach 2, our dataset contains around 38000 pseudo-labeled reviews. We then employ PMI and POS tagger in a similar way to phase 2. However, since this phase utilizes pseudo-labeled data instead of the true-label data, we set a higher threshold of 0.7 for the class label assignment.” Why is the threshold set to 0.7?
Q5: 5.1.1 “If the total polarity score of a review is below 0, we consider it as a positive prediction; if the final score is below 0, we consider it as negative;” In which case does the score below 0?
3. The least important points
Q6: Table reference error in section 3.1.2, Table3 should be changed to Table1.
Q7: 3.1.2 Incorrect data in Table1, the data in the last column of the last row in Table 1 should be 10410.
Q8: 5.1.2 “Among the three translated lexicons, VADER classifies 46.60% reviews correctly, while AFINN and Opinion Lexicon provide 48.65% and 41.95% accuracy, respectively.” The accuracy of AFINN in Table 4 is 31.65%, which does not agree with the description in the text.
Q9: CONCLUSION “We made both BengSentiLex and BengSwearLex publicly available for the researchers in Sen (2020)” This sentence lacks punctuation.

·

Basic reporting

The article has been provided in clear and professional English, as well as a sufficient organized sections. The tables and Figures are also well created.

Experimental design

The approach seems to have a promising results. Although the approaches have been used in previous papers, they showed a good insight on the specific language (Bengali).

Validity of the findings

The authors have provided well-designed conclusion, as well as defining the data with good statistics.

Additional comments

Here is a few suggestion that can improve the article:
1- The scope of the "related works" section requires a more thorough details.

i) Some essential parts of the paper, such as "supervisory characteristics" (supervised methods, semi-supervised methods, and unsupervised methods) can be discussed.

ii) the following paper has also done a similar work. it is recommended to review and compare the method with the provided method in the paper. https://www.sciencedirect.com/science/article/abs/pii/S0950705120305529

iii) Section 3.2.1, the second paragraph is relatively discussed the related works, which can be replaced in the "related works" section.

2- For better referencing the readers, it is recommended to reference the Equations with numbers.
3- For better understanding of the readers, a translation for Figure 5 is required.
4- Also, the source of the dataset (Youtube reviews) is needed to be cited.
5- The compared methods are mostly traditional methods for the experiments. authors must provide state-of-the-art methods for comparisons. (having LSTM, Bi-LSTM, CNN, Attention model, BERT model, etc.) - though not all of them, but to compare some of them.

The general idea behind the paper is a novel approach for Bengali language and hope the authors find the comments useful and make the necessary changes for the acceptance.

Reviewer 3 ·

Basic reporting

No comment

Experimental design

No comment

Validity of the findings

No comment

Additional comments

line 231, Table 3 -> Table 1
line 235, numbers do not add up to the numbers tabulated in Table 1 -> last row, last column cell?
line 249, will translation sufficiently map all types of sentiments from one to another language?
line 254, reference for NMT
line 364, which is the machine learning model used in SGD algorithm? Similary in line 515.

---

## Round 0.2 · Minor Revisions

Kindly improve the English language writing and correct the typographical mistakes.

Reviewer 3 ·

Basic reporting

no comment

Experimental design

no comment

Validity of the findings

no comment

Additional comments

The following questions should be addressed by the authors:
1) line 280 typo error
2) line 301 typo error
3) line 320, was there any statistical analysis carried out?
4) line 363, how many languages and list them?
5) line 405, what are the input features for the various ML classifiers? covered in section 5.2.2 but not in section 4!
6) line 408 and 409, svm is a binary classifier and not a multiple class classifier.
7) line 489, which type of word embedding is used, is it CBOW or count based?
8) line 529, how is the polarity score computed?

·

Basic reporting

1. The concept is clear.
2. In line no. 280, "sophisti. cated" - spelling mistake.
3. In figure 2, both Phase 2 and 3 consist of lexicon building block (you may indicate flow of phase 2 and 3 work in this diagram)
4. In line 383, 4.4.2 Phase 2: Lexicon generation from labeled data - you mentioned like that. But in figure 2, phase 2 input is taken from both labeled and unlabeled like that. Rectify this.

Experimental design

1. To improve the readability, the author requires to provide a good data flow/ sequence diagram in understanding the “Service Level Agreement”

Validity of the findings

1. In the introduction, the findings of the present research work should be compared with the recent work of the same field towards claiming the contribution made. , kindly provide several references to substantiate the claim made in the abstract (that is, provide references to other groups who do or have done research in this area).
2. Try to concise the conclusion.
3. Discuss the future plans with respect to the research state of progress and its limitations.

Additional comments

1. In the Introduction section, the drawbacks of each conventional technique should be described clearly.
2. You should emphasize the difference between other methods to clarify the position of this work further.
3. The Wide ranges of applications need to be addressed in the Introduction
4. Add the advantages of the proposed system in one quoted line for justifying the proposed approach in the Introduction section.

---

## Author Rebuttal · Round 0.2

# Reviewer 1

**Basic reporting**

no comment

**Experimental design**

The discussion of the experiment needs to be more in-depth, and the content of the experiment needs to be more detailed and complete. Please analyze the advantages of the proposed method and supplement the related experiments.

**Validity of the findings**

no comment

**Comments for the author**

1. Your most important issue

**Q1: Are the methods presented in this paper the most advanced and original, what are the main challenges of this work, and these need to be further discussed in the relevant work.**

  **-**Thank you for comments. The paper presents a new and original corpus-based approach for sentiment lexicon creation in Bengali. Till now, this is the most effective sentiment lexicon in Bengali as shown by the evaluation results.

Some of the challenges associated with this work are lack of annotated and unannotated corpus, standard part-of-speech (POS) tagger, etc. which have been included in revised manuscript**. (Line 100-106)**

Based on your comments, a new section has been added that describes challenges associated with creating this sentiment lexicon. (Line 100-106). Besides, a new section has been added to discuss how the proposed work differs from the existing sentiment lexicon creation methods in Bengali and English **(Section 2.3, Line 194-221).**

**Q2: The discussion of the experiment needs to be more in-depth, and the content of the experiment needs to be more detailed and**

**complete. Please analyze the advantages of the proposed method and supplement the related experiments.**

-Thank you for your comments. The discussion section has been extended with more information **(Line 592-609).** The created sentiment lexicon has advantage over existing Bengali lexicons as it can capture words used in informal communication.

2. The next most important item

**Q3: 3.3.2 What is the meaning of the PMI formula and its variables? Please give specific explanation.**

-Thank you for your comments. The PMI formula has been explained with the definition of various terms used in the equation. **(Line 289-293, 391-392)**

**Q4: 3.3.3 "After applying approach 2, our dataset contains around 38000 pseudo-labeled reviews. We then employ PMI and POS tagger in a similar way to phase 2. However, since this phase utilizes pseudo-labeled data instead of the true-label data, we set a higher threshold of 0.7 for the class label assignment." Why is the threshold set to 0.7?**

- Thank you for your comments. The threshold is set empirically and based on the assumption that as pseudo-labels are not perfect (unlike true label), a higher threshold need to be used compared to true label data.

Q5: 5.1.1 "If the total polarity score of as review is below 0, we consider it as a positive prediction; if the final score is below 0, we consider it as negative;" In which case does the score below 0?

- Thank you for your comment. This was a typo, 'If the total polarity score of a review is below 0, we consider it as a positive prediction', has been changed to 'If the total polarity score of a review is above 0, we consider it as a positive prediction'

3. The least important points

Q6: Table reference error in section 3.1.2, Table3 should be changed to Table1.

 - The Table reference has been fixed.

Q7: 3.1.2 Incorrect data in Table1, the data in the last column of the last row in Table 1 should be 10410.

-Thank you for pointing this. The value has been fixed.

Q8: 5.1.2 "Among the three translated lexicons, VADER classifies 46.60% reviews correctly, while AFINN and Opinion Lexicon provide 48.65% and 41.95% accuracy, respectively." The accuracy of AFINN in Table 4 is 31.65%, which does not agree with the description in the text.

- Thank you for finding this. The typo has been fixed.

Q9: CONCLUSION "We made both BengSentiLex and BengSwearLex publicly available for the researchers in Sen (2020)" This sentence lacks punctuation.

- Thank you. The punctuation has been added.

# Reviewer 2

**Basic reporting**
The article has been provided in clear and professional English, as well as a sufficient organized sections. The tables and Figures are also well created.

**Experimental design**
The approach seems to have a promising results. Although the approaches have been used in previous papers, they showed a good insight on the specific language (Bengali).

**Validity of the findings**

The authors have provided well-designed conclusion, as well as defining the data with good statistics.

**Comments for the author**

Here is a few suggestion that can improve the article:

1- The scope of the "related works" section requires a more thorough details.

i) Some essential parts of the paper, such as "supervisory characteristics" (supervised methods, semi-supervised methods, and unsupervised methods) can be discussed.

- A basic terminology section has been added that describes supervised approach, semi-supervised approach and other terms/approach used in the paper**. Section 4.1,  Line 259-293**

ii) the following paper has also done a similar work. it is recommended to review and compare the method with the provided method in the paper. https://www.sciencedirect.com/science/article/abs/pii/S0950705120305529

- The comparison with the proposed paper has been shown. **Line 160-167, Line 217-220**

**iii) Section 3.2.1, the second paragraph is relatively discussed the related works, which can be replaced in the "related works" section.**

   Thank you for your suggestion, the machine translation part has been moved to the earlier *basic terminology* section.

**2- For better referencing the readers, it is recommended to reference the Equations with numbers.**

-        Equation number has been added for better referencing.

**3- For better understanding of the readers, a translation for Figure 5 is required.**

-        Thank you for your comments. The English translation has been added based on your suggestion.

**4- Also, the source of the dataset (Youtube reviews) is needed to be cited.**

- The source of the dataset has been added.

**5- The compared methods are mostly traditional methods for the experiments. authors must provide state-of-the-art methods for comparisons. (having LSTM, Bi-LSTM, CNN, Attention model, BERT model, etc.) - though not all of them, but to compare some of them.**
- Thank you for your comments. For evaluation, I added the comparison of LSTM, Bi-LSTM and CNN with the swear lexicon for identifying profanity. **Table 5, Line 583**

The general idea behind the paper is a novel approach for Bengali language and hope the authors find the comments useful and make the necessary changes for the acceptance.

# Reviewer 3

**Basic reporting**
No comment
**Experimental design**
No comment
**Validity of the findings**
No comment
**Comments for the author**
line 231, Table 3 -> Table 1

- Thank you for your comments. The table reference number has been fixed.
line 235, numbers do not add up to the numbers tabulated in Table 1 -> last row, last column cell?

- The typo has been fixed.

line 249, will translation sufficiently map all types of sentiments from one to another language?

- Thank you for your insightful comments. In another paper, we have investigated the sentiment preservation in Bengali and Machine translated review utilizing Cohen's kappa and Gwet's AC1.  We found two very accurate classifiers, SVM and LR show kappa scores above 0.80 and AC1 scores above 0.85, which indicates sentiment consistency exists between original Bengali and machine-translated English reviews.  Thus, although not perfect, Google Machine Translation preserve the sentiment in majority of the cases.

- References to NMT has been added.

line 364, which is the machine learning model used in SGD algorithm? Similary in line 515.

- Thank you for your comments. The SGD algorithm uses hinge loss which is basically linear SVM.  The following information has been added -

    "For SGD, hinge loss and l2 penalty with a maximum iteration of 1500 are employed.'

---

## Round 0.3 · accepted · Accept

Kindly check the manuscript for minor language edits.